# Systemic Anticoagulation and Inpatient Outcomes of Pancreatic Cancer: Real-World Evidence from U.S. Nationwide Inpatient Sample

**DOI:** 10.3390/cancers15071985

**Published:** 2023-03-26

**Authors:** Yen-Min Huang, Hsuan-Jen Shih, Yi-Chan Chen, Tsan-Yu Hsieh, Che-Wei Ou, Po-Hsu Su, Shih-Ming Chen, Yun-Cong Zheng, Li-Sung Hsu

**Affiliations:** 1Hemophilia and Thrombosis Treatment Center, Division of Hematology and Oncology, Department of Internal Medicine, Keelung Chang Gung Memorial Hospital, Keelung 204, Taiwan; 8902004@cgmh.org.tw (Y.-M.H.); b9505015@cgmh.org.tw (P.-H.S.); 2Division of Hematology and Oncology, Department of Internal Medicine, Lotung Poh-Ai Hospital, Yilan 265, Taiwan; 3Institute of Medicine, Chung Shan Medical University, Taichung 402, Taiwan; 4Division of Hematology and Oncology, Department of Internal Medicine, Linkou Chang Gung Memorial Hospital, Taoyuan 333, Taiwan; samuel.shih@outlook.com (H.-J.S.); cheweiou@gmail.com (C.-W.O.); 5Department of General Surgery, Keelung Chang Gung Memorial Hospital, Keelung 204, Taiwan; a017749@gmail.com; 6Department of Pathology, Keelung Chang Gung Memorial Hospital, Keelung 204, Taiwan; 8902059@cgmh.org.tw; 7Bachelor Program in Health Care and Social Work for Indigenous Students, Providence University, Taichung 433, Taiwan; big32762992@pu.edu.tw; 8Departments of Neurosurgery, Keelung Chang Gung Memorial Hospital, Keelung 204, Taiwan; 9College of Medicine, Chang Gung University, Taoyuan 333, Taiwan; 10Department of Biomedical Engineering, National Taiwan University, Taipei 110, Taiwan; 11Department of Medical Research, Chung Shan Medical University Hospital, Taichung 402, Taiwan

**Keywords:** systemic anticoagulant, pancreatic cancer, morbidity, mortality, Nationwide Inpatient Sample (NIS)

## Abstract

**Simple Summary:**

Pancreatic cancer is known to cause a hypercoagulable state and increase the risk of thrombosis. While long-term systemic anticoagulation is a treatment option to reduce this risk, its effectiveness remains ambiguous. In this retrospective study, data from the U.S. Nationwide Inpatient Sample database were extracted and compared statistically to find the solution. The results showed that pancreatic cancer patients who had received long-term systemic anticoagulant had a lower risk of sepsis, shock, acute kidney injury, and in-hospital mortality, and a shorter hospital stay. The results suggest that long-term systemic anticoagulation should be considered as a treatment option for pancreatic cancer patients.

**Abstract:**

**Background:** Pancreatic cancer can induce a hypercoagulable state which may lead to clinically apparent thrombosis. However, the effect of anticoagulants remains ambiguous. This study aimed to investigate the potential effect of long-term systemic anticoagulant usage on hospitalization outcomes of patients with pancreatic cancer. **Methods:** This retrospective study extracted all data from the U.S. Nationwide Inpatient Sample (NIS) database from 2005 to 2018. We included hospitalized adults ≥18 years old with a pancreatic cancer diagnosis identified by International Classification of Diseases ninth revision (ICD-9) and tenth revision (ICD-10) codes. We utilized diagnostic codes ICD9 V58.61 and ICD10 Z79.01, i.e., ‘long-term use of anticoagulant’, to identify individuals who were on a long-term systemic anticoagulant. The study cohort were then further grouped as being with or without long-term systemic use of an anticoagulant. Propensity score matching was performed to balance the characteristics of the two groups. The risks of life-threatening events, e.g., acute myocardial infarction (AMI), acute heart failure (AHF), sepsis, shock, and acute kidney injury (AKI), in-hospital death, and prolonged length of stay (LOS) in the hospital were compared between the groups by univariable and multivariable logistic regression analyses. **Results:** The study population consisted of 242,903 hospitalized patients with pancreas cancer, 6.5% (*n* = 15,719) of whom were on long-term systemic anticoagulants. A multivariable regression analysis showed that long-term systemic anticoagulant use was independently associated with lower odds of sepsis (aOR: 0.81, 95% CI: 0.76–0.85), shock (aOR: 0.59, 95% CI: 0.51–0.68), AKI (aOR: 0.86, 95% CI: 0.81–0.91), in-hospital mortality (aOR: 0.65, 95% CI: 0.60–0.70), and prolonged LOS (aOR: 0.84, 95% CI: 0.80–0.89). **Conclusions:** Long-term systemic anticoagulant use is associated with better clinical outcomes in terms of decreased risks of some life-threatening events, in-hospital death, and prolonged LOS among hospitalized patients with pancreatic cancer in the U.S.

## 1. Introduction

Pancreatic cancer is the eleventh most challenging malignancy to treat, with one of the highest mortality rates out of all cancers in the United States (U.S.) [1]. It has been known as a malignant form of cancer with poor prognosis and treatments that, in most patients, ends with death. Pancreatic cancer often presents at an advanced stage, and is resistant to treatment, ranking last amongst all cancers in terms of prognosis [2]. It has a five-year survival rate of 2–9%, despite the observation that there has been a slight improvement in mortality over the past few decades due to significant advancements in the surgical and therapeutic treatment of it [3,4]. Due to the severe pain and cachexia associated with this malignancy, especially in the later stages, even implementing adequate palliative care has proven to be problematic [5,6]. In recent years, there seem to have been advances in the development of various novel treatment methods, including immune checkpoint inhibitors (ICIs), nanotechnology, polyamine metabolism, etc.; however, it has still been challenging to treat patients with pancreatic cancer up to now [7,8,9].

Venous thromboembolism (VTE) occurs in approximatively 20% of patients with pancreatic cancer, resulting in increased morbidity, mortality, and high health care costs [10]. Although the relationship between pancreatic cancer and hypercoagulability is well described, the underlying pathological mechanisms and the interplay between the proposed pathways remain matters of intensive study [10].

A recent systematic review included 35 publications on VTE prophylaxis and treatment and 18 on VTE risk assessment, suggesting that hospitalized patients with cancer and an acute medical condition need anticoagulant treatment throughout hospitalization, whereas anticoagulant treatment is not routinely recommended for all outpatients with cancer [11]. For pancreatic cancer, specifically, emerging clinical data strongly suggest that anticoagulant treatment may improve patient survival not only by decreasing the risk of thromboembolic complications but also by anticancer activity. However, the current clinical guidelines on anticoagulant applications in ambulatory patients with pancreatic cancer are ambiguous [12,13,14,15]. In this study, we utilized an extensive, nationally representative database to investigate the potential effect of long-term systemic anticoagulant usage on the hospitalization outcomes of patients with pancreatic cancer. It was hypothesized that long-term systemic anticoagulation use was associated with more favorable clinical outcomes in such a patient subgroup.

## 2. Materials and Methods

### 2.1. Study Design and Data Source

This population-based, retrospective observational study extracted all data from the U.S. Nationwide Inpatient Sample (NIS) database between 2005 and 2018. The NIS database is the largest all-payer, continuous inpatient care database in the United States, showing that there are about 8 million hospital stays each year [16]. The database is administered by the Healthcare Cost and Utilization Project (HCUP) of the U.S. National Institutes of Health (NIH). Patient data include primary and secondary diagnoses, primary and secondary procedures, admission and discharge statuses, patient demographics, expected payment sources, durations of hospital stay, and hospital characteristics (e.g., bed size, location, teaching status, and hospital region). All admitted patients were initially considered for inclusion. The continuous, annually updated NIS database derives patient data from about 1050 hospitals in 44 states in the U.S., representing a 20% stratified sample of U.S. community hospitals as defined by the American Hospital Association.

### 2.2. Ethics Statement

HCUP-NIS is a deidentified database, and the Institutional Review Board of Johns Hopkins Medical Institutions deemed that the study, using the HCUP-NIS database, is exempt from ethical review.

### 2.3. Study Population

This study utilized the International Classification of Diseases ninth revision (ICD-9) and tenth revision(ICD-10) diagnostic codes to identify adults ≥18 years old admitted to U.S. hospitals who had pancreatic cancer (ICD-9: 157.0, 157.1, 157.2, 157.3, 157.4, 157.8, and 157.9; ICD-10: C25) between 2005 and 2018 in the NIS database (ICD-9: 2005–2015Q3; ICD-10: 2015Q4–2018). Patients with no information on in-hospital death, length of stay (LOS), or with a history of other malignancies were excluded. Eligible patients were further divided into two groups according to whether they were treated with or without long-term anticoagulant use as documented in their medical records in the NIS database through ICD codes (ICD9-CM code V58.61 and ICD10-CM code Z79.01).

### 2.4. Study Variables and Outcome Measures

The study endpoints were the incidence of (1) any life-threatening events during admission; (2) in-hospital mortality; (3) unfavorable discharge, defined as transfer to nursing homes or long-term care facilities; (4) prolonged LOS, defined as LOS being ≥75th percentile. Life-threatening events included acute myocardial infarction (AMI), acute heart failure (AHF), sepsis, shock, and acute kidney injury (AKI).

### 2.5. Covariates

The patients’ characteristics included age, gender, race, household income, insurance status (primary payer), and admission type (elective or emergent). Pancreatic resection, a history of myocardial infraction (MI), a history of valvular heart disease, prior percutaneous coronary intervention (PCI)/coronary artery bypass grafting (CABG)/valve surgery, a history of coronary heart disease (CHD), a history of atrial fibrillation (AF), history of peripheral artery disease (PAD), a history of diseases of the circulatory system, a history of cerebral artery occlusion/stenosis, and long-term aspirin use were identified using ICD-9 and ICD-10 diagnostic codes. Comorbidities were also identified through ICD codes and were further graded using the Romano adaptation of Charlson’s comorbidity index (CCI) [17]. Hospital-related characteristics (bed size, location/teaching status, and hospital region) were also extracted from the database to be included as part of the comprehensive data available for all participants.

### 2.6. Statistical Analysis

The descriptive statistics of the patients were presented as unweighted counts (*n*) and weighted percentage (%) or mean ± standard error (S.E.). Comparisons between the study groups were performed using PROC SURVEYFREQ and SURVEYREG for categorical and continuous variables, respectively. Logistic regressions were performed using PROC SURVEYLOGISTIC to estimate the odds ratio (OR) and 95% confidence interval (CI) of the associations between study variables and outcomes of interest. In order to balance the baseline characteristics between the groups, the cohort was then matched into the case (with long-term systemic anticoagulant use) and control (without long-term systemic anticoagulant use) groups with a ratio of 1:3 by the propensity score using the SAS %OneToManyMTCH macro. The macro performs a 1:N case–control match on the propensity score; the analyst is allowed to specify the number of controls to match to each case. The macro makes “best” matches first and “next-best” matches next, in a hierarchical sequence until no more matches can be made [18] The propensity score assigned to each hospitalization was derived from the multivariable logistic regression model constructed to determine the likelihood of receiving a long-term systemic anticoagulant after controlling variables with a *p*-value of <0.05 in the unmatched population of the two groups. A 1:3 fixed ratio of nearest neighbor matching was performed.

Further, univariate and multivariate logistic regression models were used to calculate the odds ratio (OR) and 95% confidence interval (CI) of determining the association between study variables and the clinical outcomes. Variables that were significantly different between the study groups after matching were adjusted in the multivariable analysis. Since the data of NIS were the 20% samples of the total U.S. inpatient admissions, weighted samples (before the year 2011: TRENDWT; after 2012: DISCWT), stratum (NIS_STRATUM), and cluster analysis (HOSPID) were utilized to produce national estimates for all the analyses as instructed by the NIS guideline. A two-sided *p*-value of <0.05 was regarded as statistically significant. Data management and statistical analyses were conducted using the SAS version 9.4 software (SAS Institute, Inc., Cary, NC, USA).

## 3. Results

### 3.1. Study Population

After selection, there were 258,631 hospitalized patients aged ≥ 18 years and diagnosed with pancreatic cancer. Patients with missing data on gender and outcomes of interest (*n* = 259) were excluded. Further, 15,469 patients with a history of other malignancies were also excluded. The remaining 242,903 patients were included, of which 6.5% (*n* = 15,719) were on long-term systemic anticoagulants according to medical records. After propensity score matching, there were 48,353 patients left for subsequent analyses, representing 237,832 adults in the U.S. Amongst them, 12,087 (25.0%) patients were on long-term systemic anticoagulants whereas 36,266 (75.0%) were not (Figure 1).

### 3.2. Characteristics of Hospitalized Patients with Pancreatic Cancer, with or without Long-Term Systemic Anticoagulant Use

Table 1 shows the characteristics of the study cohort. Before matching, the mean age of the study cohort was 67.8 ± 0.05, and over half of the patients were males (50.8%). The majority of pancreatic patients were White (73.1%), had insurance covered by Medicare/Medicaid (65.9%), were admitted emergently (79.6%), with a CCI greater than 7 (31.4%), and stayed at large (64.2%), urban-teaching (63.1%) hospitals and hospitals located in the South (35.8%). The significant differences between long-term systemic anticoagulation users and non-users were based on age, sex, race, household income, insurance status, admission type, metastatic disease, cancer type, pancreatic resection, long-term aspirin use, bleeding status, all the related morbidities, CCI, hospital bed size, and region. After matching the significant unbalance variables, the frequencies of most study variables were not significantly different between the two groups, except for some: those with a history of AF (*p* = 0.002) and those with a history of diseases of the circulatory system (*p* = 0.003).

### 3.3. Clinical Outcomes of Hospitalized Patients with Pancreatic Cancer, with or without Long-Term Systemic Anticoagulant Use

Table 2 shows the clinical outcomes of hospitalized patients with pancreatic cancer before and after matching. Before matching, patients on long-term systemic anticoagulants had more AMI (2.1% vs. 1.8%), AHF (1.8% vs. 0.9%) and AKI (15.2% vs. 14.1%), but less sepsis (16.1% vs. 17.2%), shock (2.3% vs. 3.0%), in-hospital deaths (6.3% vs. 8.6%), or prolonged LOS (23.5% vs. 28.1%) (all *p*-value ≤ 0.007) compared to those without anticoagulants. After P.S. matching, a lower proportion of all events was observed among the patients with long-term systemic anticoagulants than those without, except for AHF. Long-term systemic anticoagulant use had a borderline association with AMI (yes vs. no: 2.2% vs. 2.5%, *p* = 0.053).

### 3.4. Associations between Long-Term Systemic Anticoagulant Use, Life-Threatening Events, in-Hospital Mortality, and Prolonged LOS

Table 3 shows the associations between long-term systemic anticoagulation use and the clinical outcomes of the matched cohort. After adjusting for a history of AF and history of diseases of the circulatory system, the multivariable regression demonstrated that long-term systemic anticoagulant use was significantly associated with lower risks of sepsis (aOR: 0.81, 95% CI: 0.77–0.86), shock (aOR: 0.59, 95% CI: 0.52–0.68), AKI (aOR: 0.84, 95% CI: 0.79–0.89), in-hospital mortality (aOR: 0.63, 95% CI: 0.58–0.68), and prolonged LOS (aOR: 0.82, 95% CI: 0.78–0.86). No significant association between systemic anticoagulant use and AMI (aOR: 0.88, 95% CI: 0.76–1.01) or AHF was observed (aOR: 1.01, 95% CI: 0.86–1.18).

## 4. Discussion

It has long been recognized that pancreatic cancer induces a hypercoagulable state that may lead to clinically apparent thrombosis [19]. Using the NIS database of the U.S., we found that among hospitalized patients with pancreatic cancer, long-term systemic anticoagulation use is independently and significantly associated with lower risks of some life-threatening events, including sepsis, shock, and AKI. Long-term systemic anticoagulation is also associated with a 35% decreased risk and a 16% decreased risk of in-hospital death and prolonged LOS, respectively. This is the first population-based study in the medical literature to assess the real-world associations between long-term systemic anticoagulation use and clinical outcomes during admission in patients with pancreatic cancer.

A previous analysis found that frequent and early onsets of VTE after pancreatic cancer diagnoses are associated with significantly lower PFS and OS, suggesting that studies are needed to determine whether the primary prophylaxis for VTE pancreatic cancer will improve morbidity and mortality [14]. The authors claim that their works made up the first systematic review and meta-analysis assessing the efficacy and safety of anticoagulant use in the pancreatic cancer population. Those findings partly support the present result that long-term systemic anticoagulation is associated with reduced risks of in-hospital death and prolonged LOS.

Continuously taking an anticoagulant for more than 3 months is considered long-term use of an anticoagulant [20,21,22,23]. In our cohort, 6.5% patients were on long-term systemic anticoagulants. Although VTE occurs in roughly 20% of patients with pancreatic cancer, only 6.5% patients needed or were able to use anticoagulants long-term for the purpose of VTE prevention/curation or even for its anticancer effects [24]. It was reported that patients with cancer and provoked VTE had a lower survival rate than cancer patients of a similar stage and with a similar treatment [25]. Dallos et al. [12] reported that anticoagulants reduced the rate of VTE from 10–25% to 5–10% with no impact on survival. Kakkar et al. [26] reported that the overall survival of pancreatic cancer patients receiving 5000 IU/day of delteparin for a year was 5% higher than that of patients with a placebo. Klerk et al. [27] also reported that the median survival was 15.4 months in the nadroparin group compared to 9.4 months in the placebo group after taking nadroparin for 6 weeks. In the present study, long-term anticoagulant use decreased the risk of in-hospital mortality by 16%. Thus, our conclusion that anticoagulants can reduce the risk of mortality is comparable to those of most studies.

With respect to the pathogenesis of VTE in cancers, chemotherapy exposure is considered a factor that leads to VTE in patients with pancreatic cancer and other cancers [28]. Although the mechanisms are poorly understood, cytotoxic chemotherapy damages endothelial cells, promoting clot formation, altering the expression of coagulation factors, and finally causing VTE. On the other hand, malignancy affects the hemostatic system, and the hemostatic system affects malignancy [29,30]. In particular, pancreatic cancer is one of the most prothrombotic malignancies, and this cancer-associated hypercoagulable state involves a complex interplay between platelets, coagulation factors, and key inflammatory pathways. In pancreatic cancer, the tissue factor (TF), also called CD142, appears to play a central role in promoting a prothrombotic state. Both preclinical and clinical studies demonstrate that pancreatic tumors produce high levels of TF, encoded by the F3 gene, which is secreted into the circulation in membrane vesicles. TF promotes a prothrombotic state, binding to factor VIIa, which then mediates the conversion of factor X to Xa and that of factor IX to IXa [31]. That study provided a detailed overview of the incidence and type of VTE during different phases of pancreatic cancer. It may provide a promising explanation for the reason why long-term systemic anticoagulation use reduces the risks of unfavorable clinical outcomes, as documented in the present study.

Although a consensus has emerged on anticoagulant applications following definitive pancreatectomy, in the metastatic setting, prophylactic anticoagulation use remains controversial [32,33]. The challenge relates to the difficulty of adequately capturing and measuring risk for an individual patient, making it difficult to know which patients are most likely to benefit from an anticoagulant strategy. Impaired hepatic synthetic function and renal impairment, present in many patients with metastatic disease, can also increase the risks of thrombosis and bleeding [12]. Therefore, patients with pancreatic cancer should be professionally evaluated by a physician when regularly taking a systemic anticoagulant.

The present results indicate that long-term anticoagulant use protect patients against some life-threatening events and in-hospital deaths in patients with pancreatic cancer. In clinical practice, an anticoagulant such as tinzaparin may be routinely used for patients with pancreatic cancer. A recent study demonstrated that tinzaparin enhances the antitumor effects of nab-paclitaxel and gemcitabine in mtKRAS pancreatic cancer cell lines via apoptosis in vivo. The combinatorial treatment with anticoagulant use provided extra tumor reduction by 24.3% compared to treatment with the drug only [34]. Other recent studies also suggest that anticoagulants improve survival in cancer patients through antitumor effects in addition to antithrombotic effects [35,36,37]. Although those studies do not specifically focus on pancreatic cancer, their conclusions are relevant to our finding that long-term systemic anticoagulation use was associated with better inpatient outcomes. We did not evaluate the potential synergic effect of chemotherapeutic regimens and anticoagulants due to lack of such data. Future longitudinal studies are warranted to investigate the potential antitumor effect of anticoagulants in order to improve prognosis.

The updated guideline recommends changes to previous recommendations and suggests that physicians offer an anticoagulant with apixaban, rivaroxaban, or low-molecular-weight heparin (LMWH) to high-risk outpatients with cancer. For long-term anticoagulation use at least 6 months of use is preferred because of the improved efficacy over vitamin K antagonists (VKAs). VKAs are inferior but may be used if LMWH or direct oral anticoagulants (DOACs) are not accessible. Although DOACs are easily applied, the efficiency and side-effects need to be noticed. Oral apixaban was found to have a similar anticoagulant effect to subcutaneous dalteparin in the Caravaggio study [38,39]. However, two RCTs reported on DOACs for anticoagulant use in ambulatory patients with cancer with increased risks of VTE [11]. Thus, caution with DOACs is also warranted in other settings with patients with a high risk of mucosal bleeding, and drug-drug interaction should be checked prior to using a DOAC [11]. Nevertheless, the event of bleeding was not assessed in the present study and should be included in a future investigation.

The present study is the first to evaluate the real-world associations between long-term systemic anticoagulation use and adverse inpatient clinical outcomes of pancreatic cancer. The strength of the present study is the use of a large sample representing a nationwide population. Also, this study conducted propensity score matching, defined as the conditional probability of a subject being assigned to the anticoagulation treatment group according to observed covariates, which minimized the bias by indication. This study was inherently limited by its retrospective and observational nature; therefore, the results should be interpreted carefully. Other limitations of this study were mainly related to the utilization of the ICD code system. Although important, the history of pancreatic cancer cannot be determined, and we cannot exclude the possibility that the difference in in-hospital outcomes was solely because of the difference in mortality between patients with a long-term and short-term history. Furthermore, the exact duration, dosage, type, purpose, or adherence to anticoagulants could not be assured due to a lack of ICD codes and data. There were, possibly, coding errors, as there were in other NIS studies that used the same ICD code systems. Specific regimens of anticoagulation such as low-molecular-weight heparin (LMWH) and direct oral anticoagulants could not be distinguished by the codes and thus were not analyzed. Chemotherapy-related parameters were not recorded, either. This study also lacked outpatient data and follow-up data after discharge.

## 5. Conclusions

Although there are many limitations in this NIS based study, long-term systemic anticoagulant usage is shown to provide various benefits for patients with pancreatic cancer. Prospective evidence is still warranted before integrating anticoagulation into the primary prevention strategy against adverse outcomes in these patients.

## Figures and Tables

**Figure 1 cancers-15-01985-f001:**
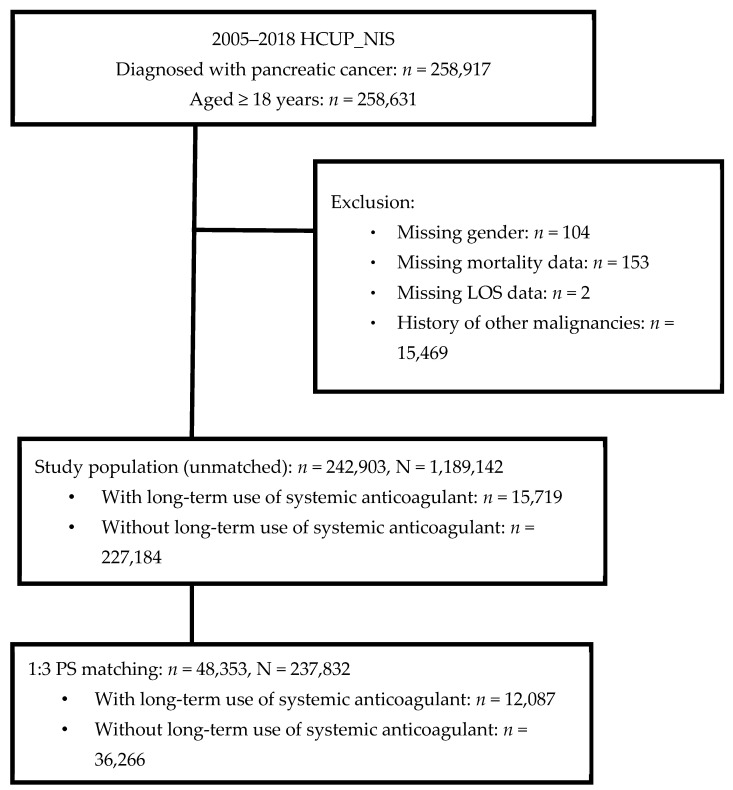
Flow diagram of study selection.

**Table 1 cancers-15-01985-t001:** Characteristics of hospitalized pancreatic cancer patients with or without long-term systemic anticoagulant use.

Characteristic	Before Matching				After Matching			
	Long-Term Systemic Anticoagulant Use				Long-Term Systemic Anticoagulant Use			
	Total (*n* = 242,903)	Yes (*n* = 15,719)	No (*n* = 227,184)	*p*-Value	Total (*n* = 48,183)	Yes (*n* = 12,045)	No (*n* = 36,138)	*p*-Value
Patients’ characteristics								
Age	67.8 ± 0.05	69.8 ± 0.11	67.6 ± 0.05	**<0.001**	70.1 ± 0.07	70.0 ± 0.12	70.1 ± 0.07	0.295
<45	7485 (3.1)	297 (1.9)	7188 (3.2)	**<0.001**	878 (1.8)	232 (1.9)	646 (1.8)	0.511
45–64	86,466 (35.6)	4609 (29.3)	81,857 (36.0)		13,935 (28.9)	3502 (29.0)	10,433 (28.9)	
65–74	73,518 (30.3)	5016 (31.9)	68,502 (30.2)		15,027 (31.2)	3789 (31.5)	11,238 (31.1)	
75–84	55,531 (22.9)	4294 (27.3)	51,237 (22.5)		13,483 (28.0)	3293 (27.4)	10,190 (28.2)	
85+	19,903 (8.2)	1503 (9.6)	18,400 (8.1)		4860 (10.1)	1229 (10.2)	3631 (10.0)	
Sex				**<0.001**				0.242
Male	119,468 (49.2)	7324 (46.6)	112,144 (49.3)		24,932 (51.8)	6290 (52.2)	18,642 (51.6)	
Female	123,435 (50.8)	8395 (53.4)	115,040 (50.7)		23,251 (48.2)	5755 (47.8)	17,496 (48.4)	
Race				**<0.001**				0.978
White	159,076 (73.1)	11,445 (78.7)	147,631 (72.7)		37,405 (77.6)	9342 (77.5)	28,063 (77.6)	
Black	28,641 (13.2)	1655 (11.4)	26,986 (13.3)		5769 (12.0)	1443 (12.0)	4326 (12.0)	
Hispanic	16,559 (7.6)	808 (5.6)	15,751 (7.7)		2794 (5.8)	708 (5.9)	2086 (5.8)	
Others	13,338 (6.1)	623 (4.3)	12,715 (6.3)		2215 (4.6)	552 (4.6)	1663 (4.6)	
Missing	25,289	1188	24,101					
Household income				**<0.001**				0.935
Quartile1	58,858 (24.7)	3425 (22.2)	55,433 (24.9)		11,061 (23.0)	2755 (22.9)	8306 (23.0)	
Quartile2	58,578 (24.6)	3802 (24.6)	54,776 (24.5)		11,579 (24.0)	2884 (23.9)	8695 (24.0)	
Quartile3	59,467 (24.9)	4041 (26.1)	55,426 (24.9)		12,117 (25.1)	3059 (25.4)	9058 (25.0)	
Quartile4	61,354 (25.8)	4205 (27.2)	57,149 (25.7)		13,426 (27.8)	3347 (27.8)	10,079 (27.9)	
Missing	4646	246	4400					
Insurance status				**<0.001**				0.359
Medicare/Medicaid	159,623 (65.9)	11,218 (71.5)	148,405 (65.5)		34,750 (72.1)	8653 (71.9)	26,097 (72.2)	
Private including HMO	71,125 (29.3)	3979 (25.3)	67,146 (29.6)		11,843 (24.6)	2970 (24.6)	8873 (24.5)	
Self-pay/no-charge/other	11,769 (4.8)	504 (3.2)	11,265 (5.0)		1590 (3.3)	422 (3.5)	1168 (3.2)	
Missing	386	18	368					
Admission type				**<0.001**				0.989
Emergent	192,576 (79.6)	13,469 (86.0)	179,107 (79.1)		41,134 (85.4)	10,282 (85.4)	30,852 (85.4)	
Elective	49,722 (20.4)	2201 (14.0)	47,521 (20.9)		7049 (14.6)	1763 (14.6)	5286 (14.6)	
Missing	605	49	556					
Metastatic disease	131,940 (54.3)	9006 (57.3)	122,934 (54.1)	**<0.001**	27,284 (56.6)	6758 (56.1)	20,526 (56.8)	0.166
Pancreatic cancer type				**<0.001**				0.450
Head	66,519 (27.4)	3374 (21.5)	63,145 (27.8)		10,659 (22.1)	2711 (22.5)	7948 (22.0)	
Body/Tail	23,694 (9.8)	1660 (10.6)	22,034 (9.7)		4714 (9.8)	1162 (9.6)	3552 (9.8)	
Islets cell	1980 (0.8)	83 (0.5)	1897 (0.8)		230 (0.5)	67 (0.6)	163 (0.5)	
unspecified	148,519 (61.2)	10,408 (66.2)	138,111 (60.8)		32,135 (66.7)	7994 (66.4)	24,141 (66.8)	
Overlapping	2191 (0.9)	194 (1.2)	1997 (0.9)		445 (0.9)	111 (0.9)	334 (0.9)	
Pancreatic resection	21,505 (8.9)	670 (4.3)	20,835 (9.2)	**<0.001**	2280 (4.7)	549 (4.6)	1731 (4.8)	0.323
Long-term aspirin use	13,859 (5.8)	1340 (8.6)	12,519 (5.6)	**<0.001**	3722 (7.8)	973 (8.1)	2749 (7.7)	0.119
Long-term antiplatelet use	1276 (0.5)	92 (0.6)	1184 (0.5)	0.301	340 (0.7)	83 (0.7)	257 (0.7)	0.783
Bleeding								
ICH	685 (0.3)	91 (0.6)	594 (0.3)	**<0.001**	196 (0.4)	51 (0.4)	145 (0.4)	0.774
Upper Gastrointestinal	10,202 (4.2)	708 (4.5)	9494 (4.2)	0.067	2806 (5.8)	715 (5.9)	2091 (5.8)	0.492
Lower Gastrointestinal	2355 (1.0)	275 (1.8)	2080 (0.9)	**<0.001**	643 (1.3)	167 (1.4)	476 (1.3)	0.594
Other	2947 (1.2)	365 (2.3)	2582 (1.1)	**<0.001**	816 (1.7)	206 (1.7)	610 (1.7)	0.850
History								
MI	9563 (3.9)	890 (5.7)	8673 (3.8)	**<0.001**	2526 (5.2)	648 (5.4)	1878 (5.2)	0.488
Valvular heart disease	7719 (3.2)	1025 (6.5)	6694 (2.9)	**<0.001**	2537 (5.3)	642 (5.3)	1895 (5.2)	0.681
Prior PCI, CABG, or valvular surgery	16,152 (6.6)	1691 (10.7)	14,461 (6.4)	**<0.001**	4481 (9.3)	1153 (9.6)	3328 (9.2)	0.255
CHD	33,556 (13.8)	2638 (16.7)	30,918 (13.6)	**<0.001**	8091 (16.7)	2006 (16.6)	6085 (16.8)	0.663
AF	27,039 (11.2)	5748 (36.6)	21,291 (9.4)	**<0.001**	17,128 (35.6)	4134 (34.4)	129,94 (36.0)	**0.002**
PAD	1680 (0.7)	200 (1.3)	1480 (0.7)	**<0.001**	457 (1.0)	122 (1.0)	335 (0.9)	0.382
Circulatory diseases	25,842 (10.7)	7037 (44.9)	18,805 (8.3)	**<0.001**	17,024 (35.4)	4397 (36.6)	12,627 (35.0)	**0.003**
Cerebral artery occlusion or stenosis	1476 (0.6)	115 (0.7)	1361 (0.6)	**0.037**	375 (0.8)	91 (0.8)	284 (0.8)	0.717
CCI				**<0.001**				0.115
0	42,788 (17.6)	2005 (12.7)	40,783 (17.9)		6252 (13.0)	1638 (13.6)	4614 (12.7)	
1–3	60,146 (24.8)	3989 (25.4)	56,157 (24.7)		12,436 (25.8)	3098 (25.7)	9338 (25.8)	
4–6	63,825 (26.2)	4136 (26.3)	59,689 (26.2)		12,379 (25.7)	3087 (25.6)	9292 (25.7)	
7+	76,144 (31.4)	5589 (35.6)	70,555 (31.1)		17,116 (35.6)	4222 (35.1)	12,894 (35.7)	
Hospitals’ characteristics								
Hospital bed size				**<0.001**				0.882
Small	31,073 (12.6)	2190 (13.8)	28,883 (12.5)		6447 (13.2)	1628 (13.4)	4819 (13.2)	
Medium	55,771 (23.2)	3716 (23.8)	52,055 (23.1)		11,703 (24.4)	2930 (24.4)	8773 (24.4)	
Large	155,326 (64.2)	9788 (62.4)	145,538 (64.3)		30,033 (62.4)	7487 (62.2)	22,546 (62.4)	
Missing	733	25	708					
Hospital location/teaching status				0.855				0.779
Rural	18,707 (7.7)	1209 (7.7)	17,498 (7.7)		3331 (6.9)	842 (7.0)	2489 (6.9)	
Urban nonteaching	71,226 (29.2)	4579 (28.9)	66,647 (29.2)		14,518 (29.9)	3595 (29.6)	10,923 (30.0)	
Urban teaching	152,237 (63.1)	9906 (63.4)	142,331 (63.1)		30,334 (63.2)	7608 (63.4)	22,726 (63.1)	
Missing	733	25	708					
Hospital region				**<0.001**				0.914
Northeast	53,941 (22.4)	3266 (21.0)	50,675 (22.5)		11,303 (23.6)	2847 (23.9)	8456 (23.6)	
Midwest	55,368 (22.8)	4284 (27.2)	51,084 (22.5)		10,095 (21.0)	2524 (21.0)	7571 (21.0)	
South	87,268 (35.8)	5043 (32.1)	82,225 (36.1)		17,177 (35.6)	4262 (35.3)	12,915 (35.7)	
West	46,326 (18.9)	3126 (19.7)	43,200 (18.9)		9608 (19.8)	2412 (19.9)	7196 (19.7)	

AF, atrial fibrillation; PAD, peripheral artery disease; CHD, coronary heart disease; ICH, intracerebral hemorrhage; MI, myocardial infarction; PCI, percutaneous coronary intervention; CABG, coronary artery bypass grafting; CCI, Charlson’s comorbidity index; HMO, health maintenance organization. Significant values are shown in bold.

**Table 2 cancers-15-01985-t002:** Inpatient outcomes of patients with pancreatic cancer with or without long-term systemic anticoagulant use.

Characteristic	Before Matching				After Matching			
	Long-Term Systemic Anticoagulation				Long-Term Systemic Anticoagulation			
	Total (*n* = 242,903)	Yes (*n* = 15,719)	No (*n* = 227,184)	*p*-Value	Total (*n* = 48,183)	Yes (*n* = 12,045)	No (*n* = 36,138)	*p*-Value
Life-threatening events	68,509 (28.3)	4687 (29.9)	63,822 (28.2)	**<0.001**	15,611 (32.5)	3611 (30.1)	12,000 (33.3)	**<0.001**
AMI	4479 (1.8)	334 (2.1)	4145 (1.8)	**0.007**	1173 (2.4)	266 (2.2)	907 (2.5)	0.053
AHF	2382 (1.0)	276 (1.8)	2106 (0.9)	**<0.001**	826 (1.7)	204 (1.7)	622 (1.7)	0.882
Sepsis	41,444 (17.1)	2527 (16.1)	38,917 (17.2)	**0.001**	9053 (18.8)	1978 (16.5)	7075 (19.6)	**<0.001**
Shock	7142 (2.9)	356 (2.3)	6786 (3.0)	**<0.001**	1569 (3.3)	260 (2.2)	1309 (3.6)	**<0.001**
AKI	34,127 (14.1)	2381 (15.2)	31,746 (14.1)	**<0.001**	8162 (17.0)	1815 (15.1)	6347 (17.7)	**<0.001**
In-hospital mortality	20,577 (8.5)	1000 (6.3)	19,577 (8.6)	**<0.001**	4285 (8.9)	761 (6.3)	3524 (9.7)	**<0.001**
Prolonged LOS ^a,b^	62,022 (27.8)	3459 (23.5)	58,563 (28.1)	**<0.001**	11,778 (26.8)	2705 (23.9)	9073 (27.8)	**<0.001**

AMI, acute myocardial infarction; AHF, acute heart failure; AKI, acute kidney injury; LOS, length of stay. ^a^: excluded in-hospital mortality patients. ^b^: length of hospital stay > 75th percentile—8 days. Significant values are shown in bold.

**Table 3 cancers-15-01985-t003:** Associations between long-term systemic anticoagulant use and inpatient outcomes of patients with pancreatic cancer after matching.

Outcomes	Long-Term Systemic Anticoagulant Use	Multivariable	
		aOR (95% CI)	*p*-Value
Life-threatening events			
AMI	Yes vs. No	0.88 (0.76, 1.01)	0.071
AHF	Yes vs. No	1.01 (0.86, 1.18)	0.894
Sepsis	Yes vs. No	**0.81 (0.77, 0.86)**	**<0.001**
Shock	Yes vs. No	**0.59 (0.52, 0.68)**	**<0.001**
AKI	Yes vs. No	**0.84 (0.79, 0.89)**	**<0.001**
In-hospital mortality	Yes vs. No	**0.63 (0.58, 0.68)**	**<0.001**
Prolonged LOS ^a,b,c^	Yes vs. No	**0.82 (0.78, 0.86)**	**<0.001**

MI, acute myocardial infarction; AHF, acute heart failure; AKI, acute kidney injury; LOS, length of stay; AF, atrial fibrillation. Significant values are shown in bold. ^a^: adjusted for history of AF and history of circulatory diseases. ^b^: excluded in-hospital mortality patients. ^c^: LOS > 75th percentile–8 days.

## Data Availability

The datasets analyzed during the current study are available from the corresponding author upon reasonable request.

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
