# Peer review of "Systemic Anticoagulation and Inpatient Outcomes of Pancreatic Cancer: Real-World Evidence from U.S. Nationwide Inpatient Sample"

_cancers, 2023, doi:10.3390/cancers15071985_

Round 1

Reviewer 1 Report (New Reviewer)

Dear Authors, 

Thank you for submitting your work here.

I have few questions:

Minor details:

Line 150: "Eligible 105 patients were further divided into two groups according to with or without long-term 106 anticoagulant use". Can I ask about the definition you use for "long" term? 0-6 months? 0-12 months?

- Any data about the pancreas cancer itself? Body vs Tail? Anticancer treatments used? Surgery?

- Line 224: "induced VTE" Do you mean provoked?

- Line 228: "delteparin" typo... Dalteparin! It is happening to me as well:)

Major Comments:

- Table 2: Can you check the numbers. I found some mistakes. As an example, for the prolonged LOS, 8980/36266=24.76% and I can read "27.3%". There are other mistakes in the table. Please check or explain why I can not find the %

-Table 2: Something is strange here. Indeed, Table 2 is reporting a higher life threatening events in patients with anticoagulation (AC) vs patients without (53.9% vs 43.2%), despite the fact that all subsequence events are more in favor of patients with AC. Why? Indeed AMI, AHF, Sepis, Shock and AKI is covering almost 95% of the population. Am I missing something?

- The Table 3 should mirror table 2 with a OR regarding life threatening events

Author Response

Reviewer 1

Line 150: "Eligible 105 patients were further divided into two groups according to with or without long-term 106 anticoagulant use". Can I ask about the definition you use for "long" term? 0-6 months? 0-12 months?

Author Response:

Thank you very much for your comment. We identified long-term use of anticoagulants through ICD codes (V58.61 for ICD9-CM and Z79.01 for ICD10-CM). Unfortunately, the definition of “long-term” is not stated in the coding system.

Several literatures may point out the definition of long-term. Patients who have had a venous thromboembolic event are generally advised to receive anticoagulant treatment for a minimum of 3 months [1]. The American College of Chest Physicians Evidence-Based Clinical Practice Guidelines suggested the periods of VTE treatment are initial (the first 7 days), long-term (7 days to 3 months), and extended (3 months onward) [2]. The European Society of Cardiology (ESC) categorize treatment as initial and long-term (3-6 months) [3]. According to the review article by Rodger et al. [4], patients taking anticoagulant for 3-6 months face the question whether to continue anticoagulants for some more time or indefinitely. Thus, we favor long term anticoagulant usage as continuing taking anticoagulant for more than 3 months. But the real answer depends on the physicians’ decision.

[1] Toth PP. Considerations for long-term anticoagulant therapy in patients with venous thromboembolism in the novel oral anticoagulant era. Vasc Health Risk Manag. 2016 Feb 10;12:23-34.

[2] Kearon C, Akl EA, Comerota AJ, Prandoni P, Bounameaux H, Goldhaber SZ, Nelson ME, Wells PS, Gould MK, Dentali F, Crowther M, Kahn SR. Antithrombotic therapy for VTE disease: Antithrombotic Therapy and Prevention of Thrombosis, 9th ed: American College of Chest Physicians Evidence-Based Clinical Practice Guidelines. Chest. 2012 Feb;141(2 Suppl):e419S-e496S.

[3] Konstantinides SV, Torbicki A, Agnelli G, et al. 2014 ESC Guidelines on the diagnosis and management of acute pulmonary embolism. Eur Heart J. 2014;35:3033–3069.

[4] Rodger MA, Le Gal G. Who should get long-term anticoagulant therapy for venous thromboembolism and with what? Blood Adv. 2018 Nov 13;2(21):3081-3087.

- Any data about the pancreas cancer itself? Body vs Tail? Anticancer treatments used? Surgery?

Author Response:
Thank you very much for your comment. The pancreatic cancer type and resection are identified by ICD codes and summarized in Table 1. Method of anticancer treatment and type of surgery cannot be determined in NIS database. We had done our best to collect useful information.

- Line 224: "induced VTE" Do you mean provoked?

Author Response:
Thank you very much for your comment. Yes, the word induced is replaced by provoked.

- Line 228: "delteparin" typo... Dalteparin! It is happening to me as well:)

Author Response:
Thank you very much for your comment. This typo is corrected.

- Table 2: Can you check the numbers. I found some mistakes. As an example, for the prolonged LOS, 8980/36266=24.76% and I can read "27.3%". There are other mistakes in the table. Please check or explain why I can not find the %

Author Response:
Thank you very much for your comment.

For the prolonged LOS a, a means Excluded In-hospital mortality patients. Therefore, 8980/(36266 total patients-3486 died in hospital )= 8980/32780=27.39%.

And because the percentage were calculated by weighted samples, stratum, and cluster, the rate of prolonged LOS is not 27.39% instead of weighted percentage of 27.3%.

The results are weighted, only the counts in Table 1&2 are unweighted. Please see the section of Statistical Analysis:

“Descriptive statistics of the patients are presented as unweighted counts (n) and weighted percentage (%) or mean ± standard error (S.E.). Comparisons between the study groups were performed using PROC SURVEYFREQ and SURVEYREG for categorical and continuous variables, respectively. Logistic regressions were performed using PROC SURVEYLOGISTIC….”

“Since the data of NIS were the 20% samples of the total U.S. inpatient admissions, weighted samples (before the year 2011: TRENDWT; after 2012: DISCWT), stratum (NIS_STRATUM), and cluster (HOSPID) were utilized to produce national estimates for all the analyses as instructed by NIS guideline.”

-Table 2: Something is strange here. Indeed, Table 2 is reporting a higher life threatening events in patients with anticoagulation (AC) vs patients without (50.9% vs 43.2%), despite the fact that all subsequence events are more in favor of patients with AC. Why? Indeed AMI, AHF, Sepis, Shock and AKI is covering almost 95% of the population. Am I missing something?

Author Response:
Thank you very much for your comment. We check the data again and revised the results in Table 2. After propensity score matching, both mortality and risk of life-threatening events are lower in patients with long-term anticoagulation application.

- The Table 3 should mirror table 2 with a OR regarding life threatening events

Author Response:
Thank you very much for your comment. After matching, the significant difference of percentages between with/ without long-term systemic anticoagulation of history of AF (p = 0.001), history of diseases of the circulatory system (p = 0.001) and long-term aspirin use (p = 0.005) were still founded (Table 1). We further conducted the multivariate analysis for the matched cohort in Table 3. And Table 2 showed the univariate analysis. Therefore, after adjusted the significant variables in Table 1, there were a little different in AMI, long-term systemic anticoagulant became borderline significant associated with AMI, while the results both showed long-term systemic anticoagulant users had the protective effects (Table 2: 2.2% vs 2.5%, p=0.042àTable 3: aOR (95% CI): 0.87 (0.76, 1.00), p=0.0501). Other life-threatening events in Table 3 had the similar result with Table 2.

Reviewer 2 Report (New Reviewer)

The abstract is clear and comprehensive. The topic was discussed in an interesting way, providing evidence for the gap of unmet need.

Line 113: abbreviations first mentioned.

The years of study is not section 2.1, should be transferred from line 152.

The years used for ICD9 and those for ICD10 were not specified.

What is the span of mortality was counted in the analysis, e.g 30 days, 60 days, within 90 days. That will be important to identify the months used for retrieving the data since the patients can not be traced across the calendar years.

The method of propensity score in statistical analysis section need to be more described. e.g the name of the package of stat tool, the variable names used as confounders, ...

The results are weighted or unweighted?

Suggest unifying font size and type in Fig 1 workflow

Table 3 MVA covariates in the model should be written somewhere either in footer or in text.

Author Response

Reviewer 2

Line 113: abbreviations first mentioned.

Author Response:
Thank you very much for your comment. The abbreviations are mentioned before usage.

The years of study is not section 2.1, should be transferred from line 152.

Author Response:
Thank you very much for your comment. The year of study is transferred from results section to section of study design.

The years used for ICD9 and those for ICD10 were not specified.

Author Response:
Thank you very much for your comment. We added the description in the Study population section. 

“Study population

This study utilized the International Classification of Diseases, Ninth Revision (ICD-9) and Tenth (ICD-10) diagnostic codes to identify adults ≥18 years old admitted to U.S. hospitals who had pancreatic cancer (ICD-9: 157.0, 157.1, 157.2, 157.3, 157.4, 157.8, 157.9; ICD-10: C25) between 2005 and 2018 in the NIS database (ICD-9: 2005-2015Q3, ICD-10: 2015Q4-2018).”

What is the span of mortality was counted in the analysis, e.g 30 days, 60 days, within 90 days. That will be important to identify the months used for retrieving the data since the patients can not be traced across the calendar years.

Author Response:
Thank you very much for your comment. Each hospitalization event in the NIS database is considered as an independent event. There is no information in the database which can be used to link different events to a patient. Thus, it is not possible to determine the span of mortality.

The method of propensity score in statistical analysis section need to be more described. e.g the name of the package of stat tool, the variable names used as confounders, ...

Author Response:
Thank you very much for your comment. We added the description in Statistical Analysis and Results section in the manuscript and as below:

Statistical Analysis:

“of 1:3 by propensity score using SAS %OneToManyMTCH macro. The macro performs a 1:N case-control match on the propensity score; the analyst is allowed to specify the number of controls to match to each case. The macro makes "best" matches first and "next-best" matches next, in a hierarchical sequence until no more matches can be made [18].”

Results:

“Table 1. shows ….The significant difference between long-term systemic anticoagulation user and non-user were founded in age, sex, race, household income, insurance status, admission type, metastatic disease, cancer type, pancreatic resection, all related morbidities, CCI, hospital bed size and region. After matching the significant unbalance variables, frequencies of most study variables…”

The results are weighted or unweighted?

Author Response:
Thank you very much for your comment. The results are weighted, only the counts in Table 1 are unweighted. Please see the section of Statistical Analysis:

“Descriptive statistics of the patients are presented as unweighted counts (n) and weighted percentage (%) or mean ± standard error (S.E.). Comparisons between the study groups were performed using PROC SURVEYFREQ and SURVEYREG for categorical and continuous variables, respectively. Logistic regressions were performed using PROC SURVEYLOGISTIC….”

“Since the data of NIS were the 20% samples of the total U.S. inpatient admissions, weighted samples (before the year 2011: TRENDWT; after 2012: DISCWT), stratum (NIS_STRATUM), and cluster (HOSPID) were utilized to produce national estimates for all the analyses as instructed by NIS guideline.”

Suggest unifying font size and type in Fig 1 workflow

Author Response:
Thank you very much for your comment. The font size and type are unified with others.

Table 3 MVA covariates in the model should be written somewhere either in footer or in text.

Author Response:
Thank you very much for your comment. To be more clear, we added the mark in Table 3, such as “Multivariable a ”and added a to the corresponding footer: “a Adjusted for history of AF, history of diseases of the circulatory system, and long-term aspirin use.”

Reviewer 3 Report (New Reviewer)

In this manuscript, Yen-Min Huang et al, perform a retrospective study ex-tracted from a US database to assess the systemic anticoagulation impact in pancreatic cancer patients. In the present study, the evolution of more than 258,000 patients with pancreatic cancer was analyzed according to whether or not they received anticoagulant treatment.

This is a very interesting study, since it is one of the first studies to analyze the effect of anticoagulant treatment as primary prevention of non-thrombotic events. One of the strengths is the large number of patients, however not all the details of the patients included in the database are available.

Overall, the manuscript is well written and important concepts are especially clear. From my point of view, before the publication of this article it is necessary to make some clarifications and changes.

General comments

-          Throughout the text, anticoagulation and thromboprophylaxis are mentioned interchangeably, in such a way that it is not possible to know if these patients were really anticoagulated at full doses or prophylactic. Please clarify this point and make the pertinent changes to the text, as it is a very important point.

-          Although it is mentioned in the text that there is a limitation of not knowing what type of anticoagulant treatment they are receiving, this point is a great limitation, since the clinical evolution is very different depending on this fact. I would like to know if they at least have the information on whether the patients received LMWH, DOAC or antivitamins K, as well as the dose of the above drugs. If not, it should be more clearly reflected in the text.

-          No mention is made in the text of the rate of bleeding presented by the patients analyzed. Since we do not have the types of treatment, I think it is essential to incorporate this information to be able to consider this article. It is essential to know that the risk of bleeding does not exceed the benefit that has been observed in the present study. Please carry out the corresponding analyzes and incorporate this information into the text.

Specific comments

- In line 360, we can read that DOACs are associated with an increased risk of bleeding, but this is not true for apixaban, according to the results of the Caravaggio trial. Please review this point and make the necessary changes.

Yours faithfully,

Author Response

Reviewer 3

Throughout the text, anticoagulation and thromboprophylaxis are mentioned interchangeably, in such a way that it is not possible to know if these patients were really anticoagulated at full doses or prophylactic. Please clarify this point and make the pertinent changes to the text, as it is a very important point.

Author Response:
Thank you very much for your comment. we modify the term “thromboprophylaxis” through out the article into either anticoagulation or anticoagulation application.

Although it is mentioned in the text that there is a limitation of not knowing what type of anticoagulant treatment they are receiving, this point is a great limitation, since the clinical evolution is very different depending on this fact. I would like to know if they at least have the information on whether the patients received LMWH, DOAC or antivitamins K, as well as the dose of the above drugs. If not, it should be more clearly reflected in the text.

Author Response:
Thank you very much for your comment. In NIS databank, the status of long-term anticoagulants usage is coded as Z79.01(ICD-10) and V58.61(ICD-9) and short-term anticoagulants usage is not even coded. There is no other code related to the anticoagulant. Thus, it is not possible to determine whether the patients received LMWH, DOAC or antivitamins K. This limitation is addressed in line 373 as “Besides, the exact duration, dosage, type, purpose, or adherence to anticoagulants cannot be assured due to lack of ICD code and data.”.

No mention is made in the text of the rate of bleeding presented by the patients analyzed. Since we do not have the types of treatment, I think it is essential to incorporate this information to be able to consider this article. It is essential to know that the risk of bleeding does not exceed the benefit that has been observed in the present study. Please carry out the corresponding analyzes and incorporate this information into the text.

Author Response:
Thank you very much for your comment. Information about the rate of bleeding presented by the patients analyzed was added in Table 1. This cohort was statistically significantly associated with bleeding variables, so propensity score matching was adjusted. Therefore, the matching values in Table 1 and Table 2 were revised.

In line 360, we can read that DOACs are associated with an increased risk of bleeding, but this is not true for apixaban, according to the results of the Caravaggio trial. Please review this point and make the necessary changes.

Author Response:
Thank you very much for your comment. The paragraph is modified and incorporate the Caravaggio study.

Reviewer 4 Report (New Reviewer)

The article by Huang YM et al. is interesting, well-written and addresses one of the most frequently asked clinical questions about the indication for prophylactic anticoagulant therapy in the pancreatic cancer patient. It is indeed correct that there are limitations in the study related to the use of the ICD-9 classification system and I would like the authors to point out this limitation in the discussion session. However the case series is large and the results interesting. 

I would ask the authors for a further effort with a summary table describing the most frequently used anticoagulants and dosages. The English is fluent and clear

Author Response

The article by Huang YM et al. is interesting, well-written and addresses one of the most frequently asked clinical questions about the indication for prophylactic anticoagulant therapy in the pancreatic cancer patient. It is indeed correct that there are limitations in the study related to the use of the ICD-9 classification system and I would like the authors to point out this limitation in the discussion session. However the case series is large and the results interesting.

Author Response:
Thank you very much for your comment. The limitation of ICD coding is stated in the last paragraph of discussion.

I would ask the authors for a further effort with a summary table describing the most frequently used anticoagulants and dosages. The English is fluent and clear

Author Response:
Thank you very much for your comment. We are willing to make a summarized table of used anticoagulants and dosage, but this cannot be done due to the limitation of NIS databank. In NIS databank, the status of long-term anticoagulants usage is coded as Z79.01(ICD-10) and V58.61(ICD-9) and short-term anticoagulants usage is not even coded. The information of anticoagulant type and dosage cannot be found in the system. Thus, although a summarized table can make the whole article important and relevant, we can only state at line 373 and stop at current place.

Round 2

Reviewer 1 Report (New Reviewer)

Thank you. No more comments

Author Response

Thank you so much for your review.

Reviewer 2 Report (New Reviewer)

Unify font in Fig1

Line 167: correct sign

Methodology: clarify if patients could be traced across calendar years or not.

Author Response

Thank you so much for your review.

Reviewer 3 Report (New Reviewer)

Dear authors,

After the changes made by the authors, I think the article can be accepted for publication.

Sincerely,

This manuscript is a resubmission of an earlier submission. The following is a list of the peer review reports and author responses from that submission.

Round 1

Reviewer 1 Report

1. what is the definition of "long-term" anti-coagulation?

2. "long-term" use of anticoagulation is basically potential bias because only long-term survivors can use long-term anticoagulant.

3. Potential readers may want to know the impact of anticoagulant on survival outcomes of pancreatic cancer in same stage.

4. Conclusion is highly predictable. 

Reviewer 2 Report

Thank you for permitting me to review the manuscript

Here are my comments 

Please elaborate which level of anticoagulation this manuscript is considering , is it preventive of curative  (full anticoagulation )? This should be discussed 

line 200 please provide reference (PPR)

Line 247 PPR I guess it is 24 

The authors have stated that: the exact duration, dosage or  adherence to anticoagulants can not be assured due to lack of data

This is a very important statement it is a very important limitation of the study in addition there is no survival comparison this should be discussed , does the lesser rate of in hospital death means lesser mortality or better outcome  in term of mortality related to the disease ? a litterature search  added to the discussion  could benefit the reader 

I think the authors should go deeper to differentiate between LMW heparin and other new anticoagulant , and at least take side which of them are better , if no they should state it that there is no difference